

# Factors influencing recurrent emergency department visits for mild acute respiratory tract infections caused by the influenza virus

Ali Cankut Tatlıparmak[1], Suleyman Alpar[2] and Sarper Yilmaz[3]

[1] Department of Emergency Medicine, Üsküdar University, İstanbul, Turkey
[2] Department of Emergency Medicine, Beykent University, İstanbul, Turkey
[3] Deparment of Emergency Medicine, Kartal Dr. Lutfi Kirdar City Hospital, İstanbul, Turkey

## ABSTRACT

**Background**. Seasonal viral outbreaks, exemplified by influenza A and B viruses, lead to spikes in emergency department (ED) visits, straining healthcare facilities. Addressing ED overcrowding has become paramount due to its implications for patient care and healthcare operations. Recurrent visits among influenza patients remain an underexplored aspect, necessitating investigation into factors influencing such revisits.

**Methods**. Conducted within a tertiary care university hospital, this study adopts an observational retrospective cohort design. The study included adult patients with acute respiratory symptoms diagnosed with influenza using rapid antigen testing. The cohort was divided into single and recurrent ED visitors based on revisits within 10 days of initial discharge. A comparative analysis was performed, evaluating demographics, laboratory parameters, and clinical process data between recurrent visitors and single visitors.

**Results**. Among 218 patients, 36.2% ($n = 139$) experienced recurrent ED visits. Age and gender disparities were not significant. Antibiotics were prescribed for 55.5% ($n = 121$) and antivirals for 92.7% ($n = 202$) of patients, with no notable influence on recurrence. Recurrent visitors exhibited lower monocyte counts, hemoglobin levels, higher PDW and P-LCR percentages, and increased anemia prevalence ($p = 0.036$, $p = 0.01$, $p = 0.004$, $p = 0.029$, $p = 0.017$, respectively). C-reactive protein (CRP) levels did not significantly affect recurrence.

**Conclusion**. This study highlights the pressing concern of recurrent ED visits among mild influenza patients, magnifying the challenges of ED overcrowding. The observed notable prescription rates of antibiotics and antivirals underscore the intricate landscape of influenza management. Diminished monocyte counts, hemoglobin levels, and altered platelet parameters signify potential markers for identifying patients at risk of recurrent visits.

Corresponding author
Ali Cankut Tatlıparmak,
alicankut@gmail.com

## INTRODUCTION

In the wake of recent pandemics such as Hemagglutinin Type 1 and Neuraminidase Type 1 (H1N1), severe acute respiratory syndrome (SARS), Middle East respiratory syndrome (MERS), and the coronavirus disease of 2019 (COVID-19), seasonal viral outbreaks often prompt patients to seek diagnosis and treatment in emergency departments (EDs) (*Abraham et al., 2016*). The phenomenon of ED crowding has progressively emerged as a public health threat, paralleling the risks posed by infectious diseases themselves (*Kenny, Chang & Hemmert, 2020*). Remarkably, in certain countries, the frequency of ED visits surpasses the nation's population, resulting in an overwhelming patient influx that strains EDs and compromises patient care (*Noel et al., 2021*).

Influenza A and B viruses lead to seasonal flu, resulting in epidemics and outbreaks globally, primarily during the fall and winter months (*Chow, Doyle & Uyeki, 2019*). Although prompt diagnosis and timely treatment initiation can alleviate disease severity, lower the risk of complications, and reduce hospital stays, certain patients might encounter a delayed amelioration of symptoms and the onset of complications, even with early diagnosis and appropriate treatment commencement (*Uyeki et al., 2019*).

Transmission, infection, and the progression of influenza virus-related diseases are intertwined with individual attributes and coexisting medical conditions, profoundly impacting the ensuing clinical trajectory of illness. Surprisingly, scant literature delves into the revisit rates of such patients (*Chow, Doyle & Uyeki, 2019*). Prompt diagnosis and timely therapeutic intervention hold paramount significance for clinical amelioration, as underscored by established guidelines (*Merekoulias et al., 2010*). Given their affordability and accessibility, complete blood count (CBC) and c-reactive protein (CRP) tests are widely employed in both initial presentations and subsequent re-visits for cases involving seasonal viral infections, encompassing EDs and primary healthcare centers. However, these factors are subject to influences from the virus, the patient, and disease characteristics (*Merekoulias et al., 2010*; *Pomorska-Mól, Markowska-Daniel & Pejsak, 2012*). Notably, influenza can prompt diverse alterations in peripheral blood parameters, warranting early consideration (*Chen et al., 2021*). However, comprehensive scientific research remains scant in exploring the interplay between these laboratory parameters, disease severity, and complications. This study seeks to fill this knowledge gap by investigating the variables that can predict recurrent visits to the ED for cases of mild influenza, providing valuable insights into a previously underexplored aspect of influenza management.

This study aims to identify the variables that can predict recurrent visits to the ED for cases of mild influenza.

## MATERIALS & METHODS

Conducted as a single-center, observational, retrospective cohort study, this research was carried out within the emergency department (ED) of a tertiary care university hospital. Ethical approval was secured from the Ethics Committee (Istanbul Medipol University Non-Interventional Ethics Committee, decision no: E-10840098-604.01.01-1308), and all research activities were in accordance with the principles outlined in the Declaration of

Helsinki. Written consent was obtained from each participating individual or their legally authorized representative in cases where the participant was deceased or otherwise unable to provide consent.

## Selection criteria

The study encompassed patients who presented at the emergency department with acute respiratory disease symptoms between June 1, 2022, and December 31, 2022. Diagnosis confirmation of influenza was performed using rapid antigen testing. Patients were included based on clinical presentation consistent with acute respiratory disease symptoms and a positive result on the rapid influenza antigen test. The inclusion criteria involved individuals aged 18 or older, diagnosed with influenza virus through rapid antigen testing, and expressing their willingness to participate. Exclusion criteria encompassed patients with SARS-CoV-2 or RSV virus infections, streptococcus infections, chronic comorbidities, pneumonia, sepsis, or multiple organ failure. Furthermore, patients who were hospitalized for any cause during or within 10 days subsequent to their ED visit, along with those whose pertinent data could not be obtained through phone communication or the Turkish Ministry of Health E-Nabiz system, were excluded from the study.

## Data collection

Patients were categorized into two distinct groups: those who initially visited the ED due to mild acute respiratory tract infection (ARI) caused by influenza (referred to as ''single visitors'') and those who had subsequent visits for mild ARI caused by influenza (referred to as ''recurrent visitors''). Laboratory profiles were evaluated based on data collected from their initial visit to the emergency department. The laboratory values obtained during the first visit were used for analysis and comparison between the recurrent and single visitor groups. Recorded data encompassed demographic information (age, gender), history of chronic comorbid diseases and associated medication usage, duration of symptoms prior to ED visit, utilization of antibiotic or antiviral (oseltamivir) medications, white blood cell count (WBC), neutrophil count, lymphocyte count, monocyte count, hemoglobin concentration, the presence of anemia, platelet count, platelet distribution width (PDW), and platelet-large cell ratio (P-LCR). Furthermore, records included occurrences of hospitalization within 10 days subsequent to the ED visit and any instances of patients being readmitted to any ED within the same timeframe after being discharged from the initial ED visit. For the purpose of this study, chronic disease was defined as a medical condition persisting for at least one year, necessitating ongoing medical assistance, or restricting daily activities. Anemia was characterized as a hemoglobin level below 12.0 g/dL in women and below 13.0 g/dL in men, following the definition set by the World Health Organization (*Cappellini & Motta, 2015*). Subsequent visits within the 10-day period were considered as potential continuations of the initial episode, given the clinical context and the likelihood of persistent symptoms rather than new, distinct episodes.

## Statistics and power analysis

Statistical analyses were executed using SPSS 28.0 for Windows (SPSS Inc., Chicago, IL, USA). The normal distribution of data was assessed through the Kolmogorov-Smirnov test

and histogram. Descriptive statistics employed numbers and percentages for categorical data, while mean ± standard deviation or median (Interquartile range (IQR) 25th–75th) was used for numerical data. Comparative analysis of continuous variables among independent groups involved the Mann-Whitney U test for data not adhering to normal distribution, and Student's $t$-test for data adhering to normal distribution. Categorical data comparison utilized Fisher or Pearson chi-square tests. The significance threshold was set at alpha < 0.05. A post-hoc power analysis was performed using the G*Power application. The power analysis was based on the observed effect size of platelet distribution width (PDW) in relation to the recurrence of ED visits. The analysis revealed a power of 0.89, with an effect size of 0.65 indicating a high level of statistical power for the observed effect size.

## RESULTS

The study comprised a total of 218 patients. Based on their revisit status to the ED with acute respiratory infection (ARI) findings within 10 days following the initial ED discharge, patients were categorized into two groups: recurrent visitors and single visitors. The rate of patient revisits was recorded at 36.2% ($n = 139$). The median age of study participants was 31 years (Interquartile range (IQR) 26–37). As presented in Table 1, no significant age difference was observed between the groups (single visitors: median age 31 (IQR 26–38), recurrent visitors: median age 31 (IQR 26–35), $p = 0.649$). Of the enrolled participants, 116 were female (53.2%) and 102 were male (46.8%). While the recurrence rate was higher among female patients (40.5%, $n = 47$) compared to male patients (31.4%, $n = 32$), this difference was not statistically significant ($p = 0.164$). The median duration of symptoms before the ED visit was 2 days (IQR 1–3), with no significant variance between the groups ($p = 0.422$). Antibiotics were prescribed for 121 patients (55.5%), and there was no significant discrepancy in the antibiotic prescription rate between the groups (single visitors: 54% $n = 75$, recurrent visitors: 58.2% $n = 46$; $p = 0.15$). Likewise, antiviral drugs (oseltamivir) were prescribed for 202 patients (92.7%), with no statistically significant distinction in the antiviral prescription rate between the groups (single visitors: 93.5% $n = 130$, recurrent visitors: 91.1% $n = 72$; $p = 0.346$).

Upon comparison of single visitors and recurrent visitors in terms of laboratory data, no statistically significant differences emerged in WBC, CRP, neutrophil count, lymphocyte count, or platelet count ($p = 0.614$, $p = 0.604$, $p = 0.929$, $p = 0.353$, $p = 0.606$, respectively). Figure 1 illustrates the difference in mean monocyte count between the recurrent visitors and single visitors groups. The mean monocyte count in the recurrent visitors group ($0.66 \pm 0.29$ $10^9$/L) was, 0.09 (95% CI [0.01–0.16]) $10^9$/L lower than that in the single visitors group ($0.74 \pm 0.29$ $10^9$/L) ($p = 0.036$). Furthermore, as depicted in Fig. 2, the PDW for the recurrent visitors group ($13.75 \pm 2.59$%) was 1.68 (95% CI [0.56–2.79]) % higher than the mean PDW of the single visitors group ($12.08 \pm 2.59$%) ($p = 0.004$). In contrast, Fig. 3 reveals that the mean P-LCR value was observed to be 3.37 (95% CI [0.34–6.4])% lower in the single visitors group ($27.71 \pm 7.86$%) compared to the recurrent visitors group ($31.08 \pm 8.67$%) ($p = 0.029$). Additionally, as indicated by Fig. 4, the mean hemoglobin

**Table 1** Analysis based on patients' recurrent admission status.

| Variables | Single visitors ($n = 139$) | Recurrent visitors ($n = 79$) | $p$ | Mean difference (95% CI) |
|---|---|---|---|---|
| Age | 31 (26–38) | 31 (26–35) | 0.649 | |
| Sex (female) | 69 (59.5%) | 47 (40.5%) | 0.164 | |
| Sex (male) | 70 (68.6%) | 32 (31.4%) | | |
| Symptom onset (days) | 2 (1–3) | 2 (1–3) | 0.422 | |
| Antibiotic presciption | 75 (54%) | 46 (58.2%) | 0.15 | |
| Antiviral prescription | 130 (93.5%) | 72 (91.1%) | 0.346 | |
| C-Reactive Protein (mg/L) | 22.56 ± 23.67 | 20.84 ± 23.3 | 0.604 | |
| White Blood Cell ($10^9$/L) | 7.47 ± 2.7 | 7.26 ± 3.28 | 0.614 | |
| Neutrophyl ($10^9$/L) | 5.39 ± 2.34 | 5.36 ± 3.07 | 0.929 | |
| Lymphocyte ($10^9$/L) | 1.18 ± 0.77 | 1.09 ± 0.61 | 0.353 | |
| Monocyte ($10^9$/L) | 0.74 ± 0.29 | 0.66 ± 0.29 | **0.036** | 0.09 (0.01–0.16) |
| Hemoglobin (g/dL) | 13.29 ± 1.54 | 12.71 ± 1.65 | **0.01** | 0.58 (0.14–1.02) |
| Anemia | 31 (22.3%) | 29 (36.7%) | **0.017** | |
| Platelet ($10^9$/L) | 225.06 ± 56.11 | 221.09 ± 51.22 | 0.606 | |
| PDW (%) | 12.08 ± 2.59 | 13.75 ± 2.59 | **0.004** | 1.68 (0.56–2.79) |
| P-LCR (%) | 27.71 ± 7.86 | 31.08 ± 8.67 | **0.029** | 3.37 (0.34–6.4) |

**Notes.**
PDW, Platelet distribution width; P-LCR, Platelet large cell ratio
Bold font indicates statistical significance.

level for the recurrent visitors group was measured at 12.71 ± 1.65 g/dL, whereas it was 13.29 ± 1.54 g/dL for the single visitors group. This signifies a mean difference of 0.58 (95% CI [0.14–1.02]) g/dL between the two groups ($p = 0.01$). Moreover, the rate of patients with anemia was significantly lower in the single visitors group (22.3%, $n = 31$) compared to the recurrent visitors group (36.7%, $n = 29$) ($p = 0.017$).

## DISCUSSION

The study revealed a concerning revisit rate of 36.2% among patients diagnosed with influenza who returned to emergency departments. Our investigation involved a comparison of demographic data, complete blood count, and CRP values between readmitted patients, aiming to discern the significance of these parameters. Our findings indicate notable statistical disparities in terms of monocyte count, hemoglobin level, PDW, and the percentage of P-LCR between the two groups. The insights gained from this study have far-reaching implications for addressing the challenges posed by recurrent ED visits among patients with mild influenza.

This study sheds light on a critical statistic: a third of patients with mild seasonal influenza who sought care at the ED experienced recurrent visits within a 10-day period. This high recurrence rate emphasizes the urgent need to develop strategies aimed at reducing non-emergency visits to emergency departments, especially for cases of mild influenza. Nonetheless, it's worth noting that the 24/7 availability of emergency departments (EDs) often leads patients to choose them due to factors such as limited access to primary health care services, concerns about trust, the perception of urgency, ease of access, advice from

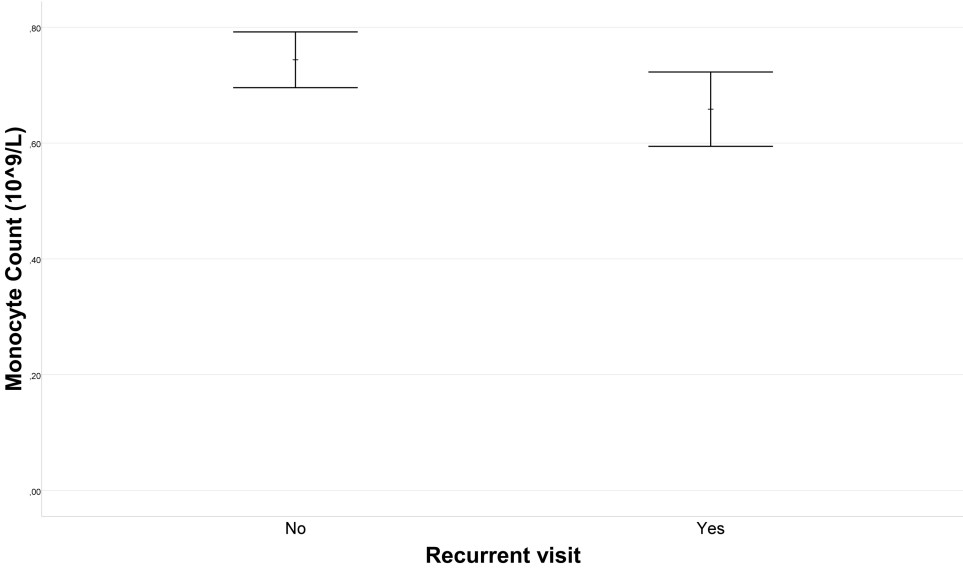

**Figure 1   Error bar plot of monocyte count in recurrent visit outcomes.** The variation in monocyte count between patients with recurrent visit outcomes (Recurrent Visitors) and those with single visit outcomes (Single Visitors). Error bars represent the standard error of the mean (SEM) for each group. The mean monocyte count was $0.66 \pm 0.29 \ 10^{9/L}$ in the recurrent visitors group and $0.74 \pm 0.29 \ 10^{9/L}$ in the single visitors group. The dashed line indicates a mean difference of $0.09 \ 10^{9/L}$, with a 95% confidence interval (CI) of 0.01–0.16 ($p = 0.036$).

friends, family, or healthcare professionals, and the belief that their condition mandates specialized healthcare services and facilities (*Coster et al., 2017*). Moreover, the context of the COVID-19 pandemic may have further influenced patients' healthcare-seeking behaviors. The heightened awareness of respiratory infections, including influenza, during the pandemic might have led individuals with mild respiratory symptoms to opt for ED visits due to concerns about the potential overlap between influenza and COVID-19 symptoms. This scenario, where patients with mild respiratory infections may have chosen the ED setting as a precautionary measure, could have contributed to the observed high recurrence rate in our study. When patients are initially diagnosed and treated in EDs for deteriorating health or unforeseen complications, they are more inclined to return at a higher rate. To alleviate ED overcrowding, it is imperative to implement regional and national measures that not only manage the influx of mild acute respiratory disease cases seeking ED care but also promote primary care utilization and non-ED healthcare services.

The investigation of recurrent visits among influenza patients has received limited attention within the existing literature. Notably, the study conducted by *Duseja et al. (2015)* offers insights into the impact of age and gender on both initial and recurrent Emergency Department (ED) visits. According to their findings, individuals aged 18–45 years exhibited a higher likelihood of ED admission compared to those over 65 years, and males displayed a greater tendency toward ED admission than females (*Duseja et al., 2015*). Conversely, a separate study indicated that elderly patients displayed an elevated probability of experiencing recurrent visits (*Schouten et al., 2022*). However, these studies

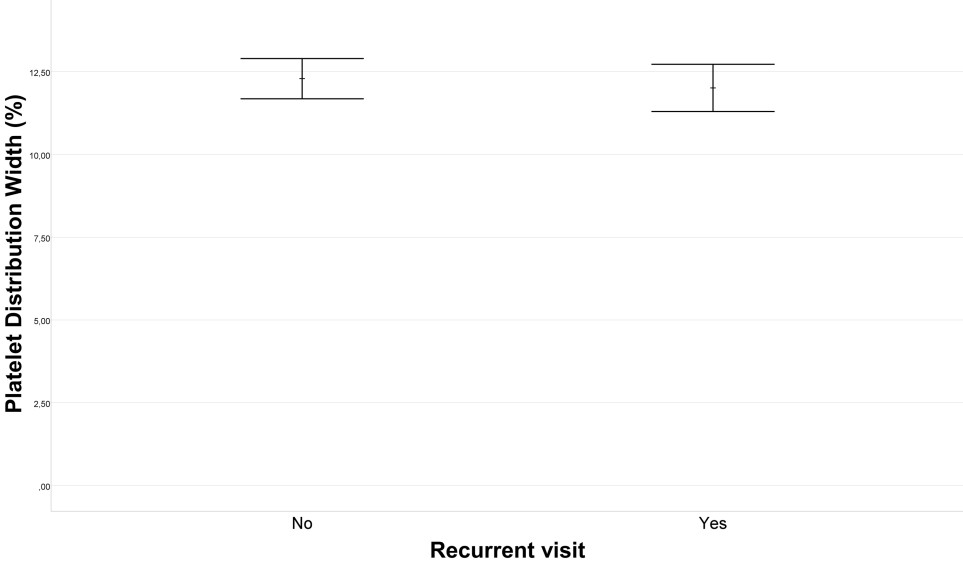

**Figure 2** **Platelet distribution width (PDW) variation in recurrent visit outcomes.** The comparison of platelet distribution width (PDW) values between patients with recurrent visit outcomes (Recurrent Visitors) and those with single visit outcomes (Single Visitors). The plot visually represents the PDW levels for each group. The PDW was $13.75 \pm 2.59\%$ in the recurrent visitors group and $12.08 \pm 2.59\%$ in the single visitors group. The difference between the groups is 1.68% (95% CI [0.56–2.79]) and is statistically significant ($p = 0.004$).

did not specifically delve into the recurrence patterns among mild influenza patients, making this research a pioneering endeavor in this realm.

Interestingly, in the context of this study, patient age did not exhibit a correlation with recurrent ED visits. Similarly, in cases of mild acute respiratory infections attributed to the influenza virus, gender did not exert a statistically significant influence on the recurrence of ED visits, although women showed a slightly higher inclination toward such revisits compared to men.

Substantial evidence supports the initiation of antiviral treatment within a two-day window from the onset of influenza virus symptoms, irrespective of vaccination status and across all age groups, to curtail complications and expedite symptom resolution (*Uyeki et al., 2019*). Within this study, two groups were analyzed, with the median onset of symptoms falling within two days of presentation, and the majority of patients commencing antiviral therapy. Upon examining the relevant literature concerning the patient demographic under scrutiny, it becomes apparent that antiviral treatment merely hastened the typical course of the disease by one day in primary care patients with influenza-like illnesses (*Butler et al., 2020*). While previous studies have explored the impact of antiviral treatment on the duration of the disease, this research extends this understanding by investigating its influence on the recurrence of ED visits, adding a novel dimension to the management of influenza cases.

Furthermore, our research delves into the complex landscape of influenza management. Clinical guidelines advocate for the assessment of bacterial co-infection in all individuals
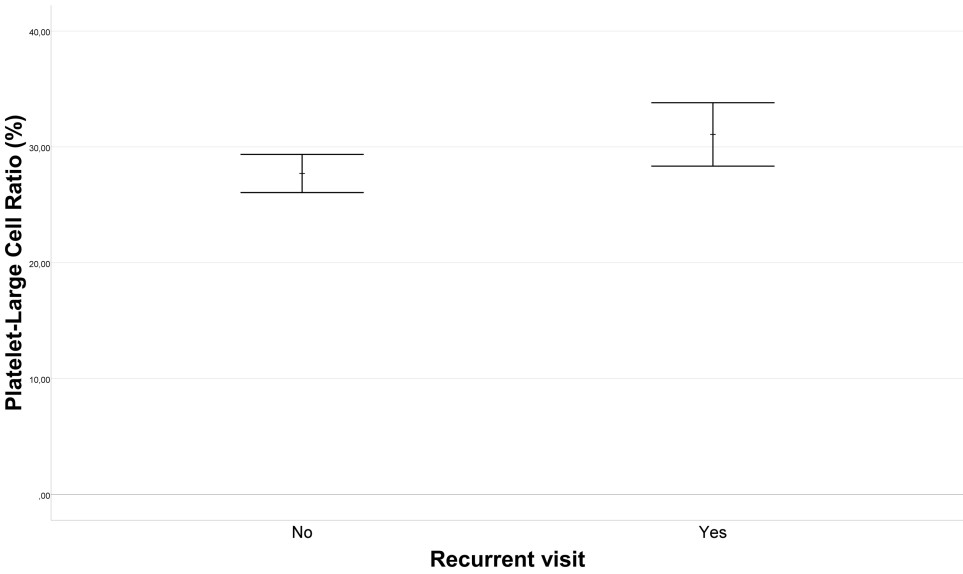

**Figure 3** **Platelet large cell ratio (P-LCR) comparison in recurrent visit outcomes.** The comparison of platelet large cell ratio (P-LCR) values between patients with recurrent visit outcomes (Recurrent Visitors) and those with single visit outcomes (Single Visitors). The graph visualizes the distribution of P-LCR levels for each group. The P-LCR was 31.08 ± 8.67% in the recurrent visitors group and 27.71 ± 7.86% in the single visitors group. This indicates a statistically significant difference of 3.37% (95% CI [0.34–6.4]) between the groups ($p = 0.029$).

suspected of or confirmed to have influenza, though routine antibiotic administration is not recommended (*Uyeki et al., 2019*). Interestingly, in this study, antibiotic prescriptions were notably prevalent in both the single visitor and recurrent visitor groups. Despite the high prevalence of antibiotic prescriptions for viral infections like influenza, our study reveals that antibiotic usage did not significantly influence the recurrence of ED visits. This finding reflects the intricate decision-making process in healthcare, particularly in the context of influenza and associated complications. While viral infections dominate, the reality of co-infections and superinfections cannot be overlooked, especially during times of heightened infectious disease concerns such as the COVID-19 pandemic. In this context, healthcare facilities have increasingly turned to empirically employing broad-spectrum antibiotics (*Ruiz-Garbajosa & Cantón, 2021*).

Monocytes and macrophages play a pivotal role in orchestrating immune responses against pathogens, notably in thwarting influenza infections (*Nikitina et al., 2018*). These cells are instrumental in curtailing influenza virus infection within lymphocytes, facilitating the synthesis and expression of viral neuraminidase, and overseeing lymphocyte apoptosis. Acute influenza infection escalates the count of circulating monocytes, and consequently, low monocyte counts are typically indicative of severe influenza infections (*Turner et al., 2020*). By identifying altered monocyte counts as a potential predictor of recurrent visits, this study delves into a novel aspect of influenza pathogenesis that has not been extensively studied before. This adds a layer of complexity to our understanding of influenza outcomes and recurrence patterns. Synthesizing these study results alongside existing literature
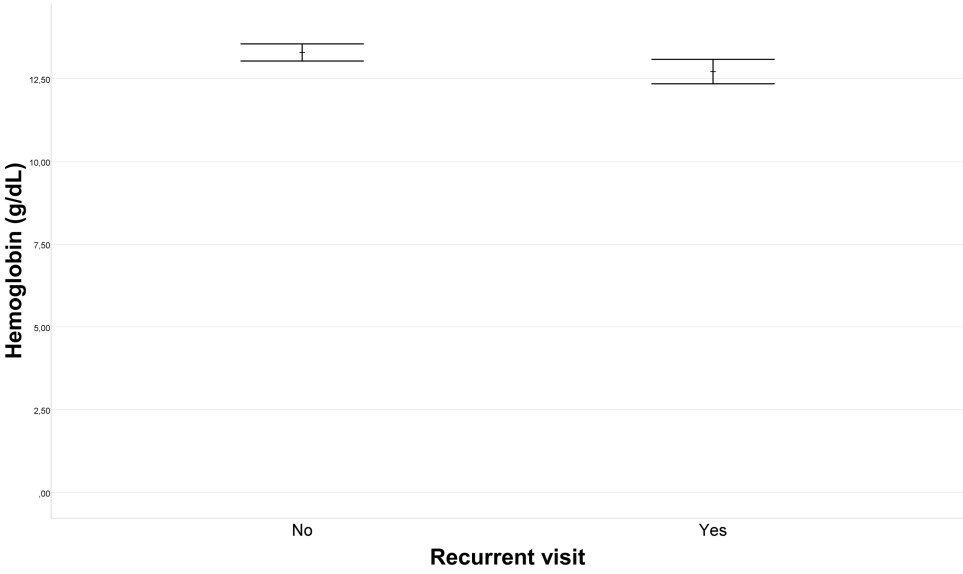

**Figure 4** **Hemoglobin level variation in recurrent visit outcomes.** The variation in hemoglobin levels between patients with recurrent visit outcomes (Recurrent Visitors) and those with single visit outcomes (Single Visitors). The plot visually presents the distribution of hemoglobin levels for each group. The mean hemoglobin level was 12.71 ± 1.65 g/dL in the recurrent visitors group and 13.29 ± 1.54 g/dL in the single visitors group. This corresponds to a mean difference of 0.58 g/dL (95% CI [0.14–1.02]) between the groups, which is statistically significant ($p = 0.01$).

suggests that monocyte reactions to influenza infections might indeed exert an influence on symptom duration and complaint resolution. Additionally, this study underscores that severe leukopenia, leukocytosis, neutrophil count, or overall white blood cell count did not impact the recurrence of presentations.

Seasonal influenza strains exhibit a heightened capacity for hemoglobin binding and agglutination compared to other influenza strains. Intriguingly, this study unveiled an association between recurrent ED visits and diminished hemoglobin levels, coupled with elevated anemia rates. Existing evidence suggests that anemia can heighten vulnerability to infections (*Oh, Song & Song, 2021*). It is worth noting that low hemoglobin levels might be attributed to heightened virus antigens due to the potent agglutination ability of the influenza virus. Given the absence of baseline hemoglobin levels, this study is confined to assessing influenza-induced alterations in hemoglobin. To ascertain whether agglutination and heightened viral loads indeed contribute to low hemoglobin in patients with recurrent ED visits, future investigations should juxtapose hemoglobin levels across visits. Such a study could potentially shed light on whether the connection between anemia and revisits is attributable to agglutination effects. The discovery of associations between altered hematological parameters and recurrent visits among influenza patients introduces a fresh perspective on influenza pathogenesis. Monocyte counts, hemoglobin levels, and platelet parameters emerge as potential markers for identifying patients at risk of recurrent visits. These findings invite further exploration into the underlying mechanisms linking these

parameters with influenza outcomes, with the potential to guide clinical practice and interventions.

Both viral and bacterial infections hold the potential to induce alterations in platelet counts (*Assinger, 2014*; *Dewitte et al., 2017*). Remarkably, within the confines of the present study, platelet counts remained within the norm for both groups, showcasing no noteworthy intergroup disparities. However, the percentages of platelet distribution width (PDW) and platelet-large cell ratio (P-LCR) emerged as significantly elevated among patients with recurrent ED visits. Notably, heightened PDW and P-LCR levels have been linked with severe acute infections of a critical nature (*Gao et al., 2014*). Notwithstanding this, investigations into the connection with influenza have been rather limited. This highlights the necessity for further research into PDW and P-LCR levels among influenza patients, aiming to elucidate their potential influence on the severity and duration of the disease.

C-reactive protein (CRP) levels can experience elevation owing to a spectrum of infectious and non-infectious conditions, precipitated by both chronic and acute inflammation. Nonetheless, their most pronounced association tends to be with infections, particularly those of bacterial origin (*Sproston & Ashworth, 2018*; *Escadafal et al., 2020*). While viral infections can also elicit CRP elevation, the degree is generally not as pronounced as in bacterial infections (*Krüger et al., 2009*). Notably, *Ziv-Baran et al. (2018)* demonstrated that even moderate elevations in CRP levels were independently correlated with emergency department visits within a seven-day timeframe. Interestingly, in our study, while the single visitor group exhibited slightly higher CRP levels in comparison to the recurrent group, CRP levels did not significantly influence the recurrence of visits. These observations underscore the need for further inquiry into the interplay between CRP levels, influenza, and the recurrence of visits, warranting additional research in this domain.

In summary, this study provides a valuable contribution to the understanding of recurrent visits among patients diagnosed with influenza. In considering the implications of our findings, it is important to acknowledge both the strengths and limitations of our study. One of the strengths lies in the comprehensive nature of our data collection, which encompassed a diverse array of demographic, clinical, and hematological variables. The large sample size further enhances the robustness of our statistical analyses and increases the generalizability of our results to a broader population of mild influenza patients. The observed high revisit rate underscores the significance of addressing non-emergency visits to EDs, particularly concerning mild influenza cases, to optimally allocate healthcare resources and mitigate ED overcrowding. The findings offer insights into the potential influence of diverse clinical and demographic variables on recurrent visits, thereby enriching the existing knowledge base on this subject matter. Moreover, the established correlation between specific hematological parameters and recurrent ED visits warrants further investigation into the underlying mechanisms and clinical ramifications. As the healthcare landscape undergoes continuous evolution, the management of challenges posed by recurrent visits necessitates a comprehensive approach that amalgamates primary care services, patient education, and healthcare policy reform. Through collaborative endeavors aimed at implementing evidence-based interventions, the establishment of an efficient and

patient-centric healthcare framework, catering both to individual patients and the larger community, becomes an attainable objective.

## LIMITATIONS

This study should be interpreted within the context of its inherent limitations. As a single-center investigation conducted in a tertiary care university hospital, the findings may have limited generalizability to broader populations or healthcare settings. The retrospective nature of the study introduces potential biases, such as incomplete data records and recall bias. The criteria used for participant selection might have influenced the composition of the study sample, potentially excluding certain patient profiles. The study's sample size, while sufficient for the analyses performed, might limit its ability to detect subtle differences in certain variables. The scope of the study did not extend to a comprehensive assessment of all potential confounding factors, which could include sociodemographic characteristics and health behaviors. Lastly, the follow-up period of 10 days might not capture all factors contributing to recurrent visits over a longer period.

## CONCLUSIONS

The findings of this study illuminated elevated rates of recurrent visits to the ED among patients grappling with mild influenza infection, coupled with a noteworthy prevalence of antibiotic prescriptions. Furthermore, it came to light that recurrent visitors exhibited diminished monocyte counts and hemoglobin levels, alongside elevated PDW and P-LCR percentages, in contrast to their first-time visitor counterparts. However, it is imperative to acknowledge that additional investigations are imperative to discern whether laboratory parameters in isolation are sufficient to elucidate the underlying causes behind this heightened revisit rate. Equally crucial is an exploration into potential involvement of other sociodemographic factors that could contribute to this phenomenon.

### Funding
The authors received no funding for this work.

### Competing Interests
The authors declare there are no competing interests.

### Author Contributions
- Ali Cankut Tatlıparmak conceived and designed the experiments, performed the experiments, analyzed the data, prepared figures and/or tables, authored or reviewed drafts of the article, and approved the final draft.
- Suleyman Alpar conceived and designed the experiments, analyzed the data, prepared figures and/or tables, authored or reviewed drafts of the article, and approved the final draft.

- Sarper Yilmaz performed the experiments, analyzed the data, prepared figures and/or tables, authored or reviewed drafts of the article, and approved the final draft.

## Human Ethics

The following information was supplied relating to ethical approvals (i.e., approving body and any reference numbers):

The Ethics Committee (Istanbul Medipol University Non-Interventional Ethics Committee, decision no: E-10840098-604.01.01-1308

## Data Availability

The raw dataset is available in the Supplementary File.

## Supplemental Information

Supplemental information for this article can be found online at http://dx.doi.org/10.7717/peerj.16198#supplemental-information.

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
