# Peer review of "Factors influencing recurrent emergency department visits for mild acute respiratory tract infections caused by the influenza virus"

_PeerJ, doi:10.7717/peerj.16198_

## Round 0.1 · original submission · Major Revisions

Dear the authors,

We have received comments from two reviewers regarding your manuscript. I hope you will be able to revise the manuscript according to reviewers' comments. I am looking forward to receiving the revised manuscript soon.

Sincerely,

Prof. Yoshinori Marunaka, M.D., Ph.D.
Academic Editor

**Language Note:** The review process has identified that the English language must be improved. PeerJ can provide language editing services - please contact us at [email protected] for pricing (be sure to provide your manuscript number and title). Alternatively, you should make your own arrangements to improve the language quality and provide details in your response letter. – PeerJ Staff

Reviewer 1 ·

Basic reporting

- The data was presented as a single table. Readers would like to see the distribution of the measurements not just point estimates. Please show distribution with plots such as bar plots or violin plots.
- Please add a section on the limitations of the study to the discussion.

- The manuscript would benefit from English editing.

Experimental design

The experimental design is appropriate for the study.

Validity of the findings

The findings are valid and consistent with the data presented.

·

Basic reporting

No comment

Experimental design

The main concerns were the study lack of novelty, and the messages were not clearly conveyed on what are the implication of the study findings on reducing non-emergency visits to emergency department (ED) and how it affects clinical practice, for example, to reduce the crowdedness of ED, besides identification of predictors.

Validity of the findings

Currently, the diagnosis confirmation of influenza was not clearly defined as in the methods section it was not stated if any PCR or other laboratory tests for confirming influenzas were done. It was not clear whether the participants had laboratory-confirmed influenza or influenza like illnesses. They authors may need to clarify this in the methods and highlights novelty in the manuscript.

Additional comments

MS TITLE: Factors influencing recurrent emergency department visits for mild acute respiratory tract infections caused by the influenza virus.

MS ID#: (#86799)

Ali Cankut Tatl1parmak Corresp., 1 , Suleyman Alpar 2 , Sarper Yilmaz 3

Thank you for the opportunity to review this paper and congratulations to the authors for completing their study. The study evaluated the factors influencing recurrent visits to the ED in patients with mild acute respiratory infections (ARI) caused by infuenza and reported that Monocyte counts, and haemoglobin concentrations was lower in participants with recurrent visits.

The main concerns were the study lack of novelty, and the messages were not clearly conveyed on what are the implication of the study findings on reducing non-emergency visits to emergency department (ED) and how it affects clinical practice, for example, to reduce the crowdedness of ED, besides identification of predictors. Currently, the diagnosis confirmation of influenza was not clearly defined as in the methods section it was not stated if any PCR or other laboratory tests for confirming influenzas were done. It was not clear whether the participants had laboratory-confirmed influenza or influenza like illnesses. They authors may need to clarify this in the methods and highlights novelty in the manuscript.

Here are some feedbacks that were structured based on the articles section.

1. Abstract
a. Currently it is not quite clear whether the study evaluated factors that cause recurrent ED visit patient with Influenza or treatment received. From my understanding the outcome will be history of recurrent ED visits (yes/no) among patients with mild influenza virus infections and the exposure variables will be some demographic and lab profiles.
b. The conclusion stated that its antibiotics prescription is high although it was not significantly different between recurrent vs single ED visit patients. This perhaps need clarification.
c. The current conclusion was just summarizing results. The conveyed messages were not clear as well as the clinical implications as above.

2. Methods section
a. Since the study mentioned acute respiratory tract infections (ARI) caused by influenza virus, what laboratory methodology was performed to confirm Influenza? If this was not confirmed, perhaps just stated as ARI.
b. It was stated that the method is retrospective cohort but there was a statement that “Written consent is obtained from each participant” (line 78). Could the authors please clarify?
c. How the sampling methods and how the samples size was calculated?
d. For those with recurrent visit, which laboratory profiles were evaluated? The first or the subsequent visit?
e. How to make a distinction that the subsequent visit(s) was not continuation of the earlier episode that has not completely cured?

3. Result section
a. The order of reporting mean and OR were not standard. The mean
135 monocyte count in the recurrent visitors group (0.66±0.29 109/L) was 0.09 (95% CI 0.01 - 0.16) 109/L lower than in the single visitors group (0.74±0.29 109/L) (p=0.036) (line 134-136). Please make adjustment as per the journal standard.

4. Discussion section
a. Discussion section has not addressed the main issues such as what are the clinical implications that support the study aim and how then we applied this predictor to prevent non-emergency visits to ED.
b. The current study did not explore the reasons why participants chose hospitals over primary health care service so perhaps paragraph 2 need more supporting data (line 152-162). The study was conducting in the COVID-19 pandemic era so perhaps this was one of the driven reasons for people with mild respiratory infection to visit ED.
c. Some of the non-significant findings were discussed in the discussion’s sections (age, gender, antiviral and antibiotics prescription).
d. Strength and weaknesses of the study were not discussed in the discussion section.

Thank you very much and hope these comments are helpful.

---

## Round 0.2 · accepted · Accept

Dear the author,

Two original reviewers have not responded to my invitation to re-evaluate your revised manuscript. This indicates the reviewers either do not want to re-review, or are happy for the academic editor to go ahead and make a decision without their further input. Therefore, I believe that your manuscript has been appropriately revised in response to the reviewers' original comments. Thus, I am pleased to inform you that your manuscript has been accepted for publication in PeerJ.

Sincerely,

Prof. Yoshinori Marunaka, M.D., Ph.D.
Academic Editor, PeerJ